

# Contrail cirrus radiative forcing for future air traffic

Lisa Bock[1] and Ulrike Burkhardt[1]

[1]Deutsches Zentrum für Luft- und Raumfahrt (DLR), Institut für Physik der Atmosphäre, Oberpfaffenhofen, Germany

*Correspondence to*: Lisa Bock (lisa.bock@dlr.de)

**Abstract.** The climate impact of air traffic is to a large degree caused by changes in cirrus cloudiness resulting from the formation of contrails. Contrail cirrus radiative forcing is expected to increase significantly with time due to the large projected increases in air traffic. We use ECHAM5-CCMod, an atmospheric climate model with an online contrail cirrus parameterization including a microphysical two-moment scheme, to investigate the climate impact of contrail cirrus for the year 2050. We take into account the predicted increase in air traffic volume and changes in propulsion efficiency and emissions, in particular soot emissions, and the modification of the contrail cirrus climate impact due to anthropogenic climate change.

Contrail cirrus radiative forcing increases by a factor of 3 from 2006 to 2050, resulting from the increase in air traffic volume and a slight shift of air traffic towards higher altitudes. Large increases in contrail cirrus radiative forcing are expected over all of the main air traffic areas but relative increases are largest over South-East Asia/India and Eastern China/Japan. The projected upward shift of air traffic attenuates contrail cirrus radiative forcing increases in the mid-latitudes but reinforces it in the tropical areas. Climate change has an insignificant impact on global contrail cirrus radiative forcing. Of the emission reductions it is the soot number emission reductions by 50% that lead to a significant decrease in contrail cirrus optical depth and coverage, leading to a decrease in radiative forcing by approximately 15%. The strong increase in contrail cirrus radiative forcing due to the projected increase in air traffic volume cannot be compensated for by the decrease in initial ice crystal numbers due to reduced soot emissions and by improvements in propulsion efficiency.

## 1 Introduction

Air traffic has contributed approximately 5% to the anthropogenic climate forcing in 2005 (Lee et al., 2009), and its contribution is rising due to the large yearly increases in air traffic (ICAO, 2007). Radiative forcing due to contrail cirrus, consisting of linear contrails and the cirrus clouds arising from them, is the largest known radiative forcing component associated with air traffic, larger than that due to $CO_2$ accumulated from aviation (Burkhardt and Kärcher, 2011). Contrail cirrus are central for mitigation efforts due to their short live time by, for example, varying flight height, path or timing or using alternative fuels, new engine designs or other technological advances (e.g. Noppel and Singh, 2007; Lee et al., 2010; Newinger and Burkhardt, 2012; Deuber et al., 2013; Burkhardt et al., 2018). Both, their large climate impact and their suitability for mitigation underline the importance of investigating contrail cirrus for future air traffic scenarios.



The climate impact of contrail cirrus in the future is determined by a number of factors: the strength and geographic distribution of the increase in air traffic volume, improved fuel efficiency, changes in aircraft emissions when using alternative fuels and the change in the background atmospheric state due to future climate change. Several projections for future air traffic volume and its emissions exist. According to ICAO (2007) and Airbus (2007) passenger flown distance is

expected to double roughly every 15 years. The air traffic inventory AEDT (Wilkerson et al., 2010) estimates that in 2050 the air traffic volume will have quadrupled relative to the year 2006. The distribution of air traffic as well as its future increase is globally very uneven. In 2006 93% of aviation fuel was burned in the Northern Hemisphere and 69% between $30°$ and $60°N$. More than half of global aviation $CO_2$ is emitted over three regions: the United States (26%), Europe (15%) and East Asia (11%) (Wilkerson et al., 2010). Due to historically low air traffic densities in the tropics, the relative increases

are expected to be much larger in the tropical areas than in the extratropics.

Lee et al. (2009) estimate that fuel usage is expected to increase between 2000 and 2050 by a factor of 2.7 to 3.9, depending on the IPCC SRES scenario while AEDT estimates an increase by a factor of 2.7 to 5 between 2006 and 2050 (Chen and Gettleman, 2016). Aerodynamic changes, weight reductions, more fuel efficient engines and an increased operational efficiency lead to increased overall fuel efficiency (Lee et al., 2009). ICAO (2007) expects a fuel efficiency improvement of

2% per annum until 2050. Increasing fuel efficiency of engines leads to an increase in the contrail formation probability and contrail radiative forcing (Marquart et al., 2003).

Measurements behind aircraft (Beyersdorf et al., 2014; Moore et al., 2017) show that the combustion of alternative fuel, a blend of Jet A and Fisher-Tropsch fuel, induces a decrease in the mass and number of soot particles. This results in a lower number of nucleated ice crystals (Kärcher and Yu, 2009; Kärcher et al., 2015) and in a higher survival rate of ice crystals

during the contrail's vortex phase (Unterstrasser, 2016). The change in the ice crystal number after the vortex phase has an impact on the microphysical process rates and the evolution of contrail cirrus (Bier et al., 2017) decreasing the mean optical depth and life time of contrail cirrus (Burkhardt et al., 2018). This is particularly important in large scale and long lived contrail cirrus clusters (Bier et al., 2017), which are responsible for a large part of the contrail cirrus radiative forcing (Burkhardt et al., 2018).

With climate change caused by increasing greenhouse gas concentrations, contrail cirrus formation and properties may change. The increase in temperature may lead to a lower contrail formation probability in particular in the tropics and in summer in the subtropics (Marquart et al., 2003). An increase in atmospheric water vapor concentration may lead to higher contrail cirrus ice water content and optical depths. A decrease in the ice supersaturation frequency (Irvine et al., 2015) may result in lower contrail cirrus coverage and associated radiative forcing.

The radiative forcing of line-shaped contrails (the contrails that have retained their initial line-shape and are, therefore, easily distinguishable from natural clouds in satellite images) and contrail cirrus for the year 2050 has been studied in a number of publications. Minnis et al. (1999) estimate a radiative forcing due to line-shaped contrails for the year 2050 of 100 mWm$^{-2}$ when assuming a constant optical depth of 0.3 and 60 mWm$^{-2}$ for varying optical depth. In Marquart et al. (2003) line-shaped contrail radiative forcing increases from 2015 to 2050 by a factor of approximately 1.6 amounting to 15 mWm$^{-2}$ in the year



2050 or after a suitable correction for a low bias in optical depth to about 45 mWm$^{-2}$ (Kärcher et al., 2010). For contrail cirrus, comprising of line-shaped contrails and the clouds developing from them, Lee et al. (2009) scaled present-day radiative forcing estimates, from models and observations, to 2050 arriving at a range between 27 and 315 mWm$^{-2}$ with no best estimate given. Chen and Gettelman (2016) studied the change in cirrus cloudiness due to contrail formation using a

model in which contrail formation is treated as a source term for cirrus ice crystals and the model microphysics is applied to a mix of contrail and natural cirrus ice crystals. They estimated that contrail cirrus radiative forcing increased by a factor of 7 from 2006 to 2050, reaching 87 mWm$^{-2}$ in the year 2050, a factor that is approximately double the factor of increase in air traffic volume. They argued that this is caused by the non-uniform regional increase in air traffic and different sensitivities of contrail cirrus radiative forcing in different regions.

Our aim is to estimate contrail cirrus radiative forcing for the year 2050 globally and regionally, isolating changes due to the increase and upward shift in air traffic volume, due to climate change and due to changes caused by the use of alternative fuels and changes in the propulsion efficiency. We use the atmospheric general circulation model coupled with a contrail cirrus scheme, ECHAM5-CCMod (Bock and Burkhardt, 2016a; Sect. 2.1), which treats contrail cirrus as an independent cloud class. The model simulates the whole life cycle of contrail cirrus and resolves the competition of the two cloud classes,

natural clouds and aircraft induced clouds, for water vapor. We apply ECHAM5-CCMod to future aviation emission scenarios from the AEDT inventory (Sect. 2.2) and estimate contrail cirrus coverage, optical depth and radiative forcing for air traffic for the year 2050 (Sect. 3). Discussion and conclusions are given in Sect. 4 and 5.

## 2. Model and data

### 2.1 CCMod in ECHAM5

We use a contrail cirrus scheme developed for ECHAM5 (Bock et al., 2016a) which is based on the contrail scheme of Burkhardt et al. (2009) and the two-moment microphysical scheme of Lohmann et al. (2008). The scheme introduces a new cloud class, contrail cirrus, in the ECHAM5-HAM model (Roeckner et al., 2003; Stier et al., 2005) with contrail cirrus modifying the atmospheric heat and water budget, thus feeding back on natural clouds (Burkhardt et al., 2011). The prognostic variables in the parameterization are contrail cirrus cover, volume and length, ice water content and ice crystal

number concentration. Contrail cirrus properties change due to the following parameterized processes: contrail formation, contrail cirrus volume growth due to turbulent diffusion and sedimentation, contrail spreading due to vertical wind shear, water vapor deposition and sublimation on contrail ice crystals, contrail ice crystal sedimentation and precipitation and indirectly due to contrail induced changes in the diabatic heating rates. Contrail cirrus form according to the Schmidt-Appleman criterion (Schumann et al., 1996) and persist in ice supersaturated regions which are parameterized in the model

(Burkhardt et al., 2008; Lamquin et al., 2012). Contrail cirrus are initialized with the air traffic density (distance per grid box) and water vapor emissions prescribed by an air traffic inventory and with an ice crystal number concentration and a contrail cross sectional area inferred from observations (Bock and Burkhardt, 2016a). If persistent, contrails spread and





accumulate more ice from ambient water vapor as long as supersaturation prevails. Contrail cirrus gradually vanish through ice crystal sedimentation into subsaturated areas and through sublimation. Hence, the whole life cycle of contrail cirrus is simulated.

We calculate total contrail cirrus coverage assuming a maximum random overlap of contrail cirrus in the vertical for each

column (Burkhardt and Kärcher, 2011). This implies that contrail cirrus coverage above or below other cirrus overlaps maximally, whereas contrail cirrus that is vertically separated from other cirrus by cloud free air overlaps randomly. We estimate the stratosphere adjusted radiative forcing, that is the change in the radiation flux at the top of the atmosphere, after the stratosphere has reached a new radiative balance (Hansen et al., 1997).

## 2.2 Inventory

The gridded aviation emissions database, developed at the Volpe National Transportation Center using the U.S. Federal Aviation Administration Aviation Environmental Design Tool (AEDT) (Roof et al., 2007; Barret et al., 2010), is composed of one base case for the year 2006 (**2006**), which has been compared against other aviation emissions data sets (Wilkerson et al., 2010), and two future 2050 scenarios. The latter include the projected increase in air traffic (**2050 Baseline**), which is based on the IPCC FESG (Forecasting and Economic Analysis Sub-Group) consensus demand forecast (FESG, 1998), and

additionally an increase in the propulsion efficiency (**2050 Scenario 1**). We use inventory data of air traffic density (distance per grid box) and water vapor emissions to initialize contrails in the model. The flight path distance for 2050 Baseline and 2050 Scenario 1 is only provided as monthly mean aggregated ground projected path distance per grid cell (track distance). Therefore, we could not use the 3D flight path distance per grid cell (slant distance) as in Bock and Burkhardt (2016b), which results in an underestimation of the initial volume and ice crystal number of contrails and, therefore, of the total

contrail cirrus radiative forcing (Bock and Burkhardt, 2016b).

Flight distance is expected to increase between 2006 and 2050 by approximately a factor of 4 (Table 1). Due to changes in aviation technologies flight altitudes are expected to shift slightly upwards. In 2006 air traffic is largest at about 240 hPa, whereas in 2050 air traffic is predicted to be largest at about 200 hPa (Fig. 1a). The regional distribution of air traffic for 2050 is expected to remain close to the distribution for 2006 with main air traffic maxima over Europe and the U.S. (Fig. 1c).

Additionally to those air traffic maxima, air traffic over Eastern and Southeastern Asia is strongly increased. Maxima in zonal mean aircraft density remain between 30 and 50°N (Fig. 1b).

## 2.3 Simulation setup

We have performed the following simulations:

- a control simulation for the air traffic of 2006 (simulation **2006**);

- a simulation with increased air traffic according to the AEDT projection of air traffic for the year 2050 (simulation **2006 Plus**);

- a simulation that additionally accounts for a changed background climate in 2050 (simulation **2050 Baseline**);



- a simulation that considers additionally an increase in fuel and propulsion efficiency as well as a change in emissions connected with the use of renewable alternative fuel, in particular a reduction in soot emissions by 50%, and a slight increase in the water emission coefficient connected with the use of alternative fuels (simulation **2050 Scenario 1**).

The specifications for the different simulations are summarized in Table 1.

All simulations were performed over 5 years with ECHAM5-CCMod at T42L41 resolution with a time step of 15 minutes. The $CO_2$ mixing ratio is prescribed for the respective base year (381ppm for the year 2006 and 478ppm for the year 2050 following the RCP 6.0 (Meinshausen et al., 2011)). The annual cycle of sea surface temperature and sea ice concentration were taken from the AMIP II database for the year 2006 and from simulations with the Hadley Centre Global Environment Model version 2 - Earth System (HadGEM2-ES) Model (Jones et al., 2011) following the Representative Concentration

Pathway (RCP) 6.0 for the year 2050. In order to calculate the contrail formation criterion we prescribe the emission index of water vapor to be 1.21 kg-$H_2O$ per kg-fuel and the combustion heat $43*10^6$MJ/kg (Chen et al., 2012). The radiation scheme is called every half hour calculating radiative transfer with and without contrail cirrus (see Bock and Burkhardt (2016b) for details).

Using alternative aviation fuels reduces soot emissions in terms of mass as well as of particle number (Moore et al., 2015,

2017). This in turn leads to a reduction in ice crystal nucleation within contrails (Kärcher et al., 2015) and to a reduction in the ice crystal loss in the vortex phase (Unterstrasser, 2016). Additionally, using alternative fuels causes a slight increase of the water emission coefficient by 15% (Moore et al., 2017). In our study we initialize contrails at a contrail age of ~7 min with a contrail cross sectional area of 200×200 m and an ice crystal number concentration of 150 cm$^{-3}$, a value derived from in situ measurements of young contrails after the vortex phase (Febvre et al., 2009; Schröder et al., 2000; Voigt et al., 2011;

Bock and Burkhardt, 2016), neglecting the variability due to the influence of the atmospheric state on ice crystal nucleation and ice crystal loss within the contrail's vortex phase. In simulation 2050 Scenario 1 we assume that a 50% reduction in soot emissions causes a 50% reduction in the initial ice crystal number.

We analyze the change in contrail cirrus properties in different areas defining four equally sized regions of high air traffic density, the U.S./Mexico (20-45°N, 235-290°O), Europe (35-70°N, -20-35°O), South East Asia/India (-10-20°N, 70-110°O)

and Eastern China/Japan (20-45°N, 95-150°O). Additionally two latitude bands (with different areal coverage) representing different background climate conditions are compared, the tropics (0-30°N) and midlatitudes (40-70°N) (see Fig. 1c and d).

## 3. Results

In this section we describe the change in simulated contrail cirrus properties and radiative forcing prescribing air traffic for the years 2006 and 2050. We distinguish between changes resulting only from the increase in air traffic and its upward shift,

and from increasing air traffic within a changed climate state. Finally we discuss an additional change in propulsion efficiency and aircraft emissions.



## 3.1 Air traffic for the year 2006

Our simulation for the year 2006 which we use as a reference has already been described in detail in Bock and Burkhardt (2016b). Differences between the simulation presented here and in Bock and Burkhardt (2016b) are due to the fact that we use here track distance as a measure for aircraft flight movements (Sect. 2.2). Using slant distance instead of track distance

leads to an increase in global air traffic volume by 1.8 with increases being largest at lower levels and over Europe and northern America. The radiative forcing due to air traffic in the year 2006 using the ground projected estimate for air traffic distance amounts to approximately 49 mWm$^{-2}$ (Fig. 2; see also Bock and Burkhardt, 2016b, Table 1), whereas the radiative forcing estimate using slant distance is larger by a factor of 1.14 (Bock and Burkhardt, 2016b).

Of the four equally-sized air traffic areas indicated in Fig. 1c and Fig. 2, flight density is largest over the U.S./Mexico and
second largest over Europe for the year 2006 (Fig. 3a). Consistently, the maxima of contrail cirrus coverage are over U.S./Mexico and Europe (Fig. 4 d) and the contribution to global contrail cirrus radiative forcing is largest from these two regions and amounts to 27% and 18%, respectively (Fig. 3b). Contrail cirrus radiative forcing per flight distance is significantly larger over Europe than over the U.S./Mexico, although optical depth is larger over the U.S./Mexico (Fig. 4 e and f). This is in agreement with the fact that a large portion of the contrail cirrus coverage over Europe is due to aged
contrail cirrus reinforced by contrail cirrus transported into Europe from the Atlantic air traffic corridor. The contribution of contrail cirrus radiative forcing from the South East Asia/India region to global mean radiative forcing is low (Fig. 3b), about 5%, but relative to the air traffic distance flown in the area very high (Fig. 3c). In this area the ice supersaturation frequency is very high (Lamquin et al., 2012) leading to a high probability of contrail formation and the amount of water vapor available for deposition is large leading to a high optical depth (Fig. 4e and f; Bock and Burkhardt, 2016b). It is important to
point out that contrail optical depth is to a large degree controlled by the number of ice crystals formed in the contrail (Burkhardt et al., 2018, Bier et al., 2017) and that this number may be reduced in the tropics due to contrail formation close to the formation threshold (Kärcher et al., 2015) leading to lower optical depth than estimated here. On average, ice supersaturation frequencies and contrail cirrus radiative forcing is in the whole tropical belt smaller than over South East Asia/India.

Contrail cirrus in the tropics are estimated to have a smaller radiative impact, absolute and per flight distance, than in the mid-latitudes (Fig. 3b and c). Ice supersaturation frequencies are on average smaller than over South-East Asia. The radiative impact per contrail coverage (not shown) is in the tropics larger than in the extratropics due to the larger optical depth than in the extratropics (Fig. 4e and f).

## 3.2 Increased air traffic

The increase in global air traffic volume, including the shift to higher altitudes (Sect. 2.2), leads to a large increase in contrail cirrus radiative forcing (Table 1, Fig. 2). While the global flight distance increases from 2006 to 2050 by a factor of about 4, the global radiative forcing increases from 49 to 159 mWm$^{-2}$, by a factor of about 3. The global pattern of contrail cirrus



radiative forcing changes only slightly, with maxima over eastern and south-eastern Asia gaining in importance (Fig. 2). Spatial differences in the increase of contrail cirrus radiative forcing are largely due to the unequal global distribution of the increase in flight distance and due to differences in the response to shifting air traffic to higher altitudes and due to saturation effects.

The shift in air traffic to higher altitudes leads in the mid-latitudes to a shift of a large fraction of air traffic into the stratosphere, where fewer persistent contrails can form due to the lower atmospheric humidity. Therefore, the increase in radiative forcing is substantially smaller than in flight distance, leading to a strong decrease (~37%) in contrail cirrus radiative forcing per flight distance in the mid-latitudes (Fig. 3c). This decrease is most pronounced over Europe (amounting to ~48%), our most northern analyzed area. Over the U.S./Mexico and Eastern China/Japan, radiative forcing per flight

distance decreases similarly, by about 30%.

In the tropics, the upwards shift of air traffic leads to a larger probability of contrail formation. Contrail formation at lower air traffic altitudes in the tropics is mostly limited by temperature which is too high for contrail formation (Burkhardt et al., 2008). Shifting air traffic in the tropical troposphere upwards, towards lower temperature conditions, thus leads to a higher probability of contrail formation. This change in contrail formation probability together with the increase in flight distance

leads to a large increase in contrail cirrus radiative forcing (Fig. 3b). The radiative forcing per flight distance decreases slightly but remains larger in South East Asia/India than in all other areas (Fig. 3c). The largest relative increase in flight distance and contrail cirrus radiative forcing is expected in the region of East China/Japan and Southeast Asia/India (Fig. 4a) but their absolute contribution to global contrail cirrus radiative forcing still remains far smaller than those from the U.S./Mexico and from Europe (Fig. 3b).

**3.3 Climate change**

We calculate contrail cirrus properties and radiative forcing for air traffic for the year 2050 within a warming climate in our 2050 Baseline simulation. The background meteorology in 2050 is assumed to change according to the RCP 6.0 scenario. The RCP scenario does not include the climate impact of contrail cirrus. In a changed climate we estimate contrail cirrus radiative forcing to amount to 160 mWm$^{-2}$ (Table 1). The net impact of climate change on global contrail cirrus radiative

forcing for the year 2050 is not significantly different from zero.

Figure 5a shows the zonal mean changes in contrail formation frequency from 2006 to 2050 meteorology. Whereas north of about 30° to 40°N the contrail formation frequency increases above 250hPa, it decreases in the tropical regions between 100 and 300 hPa. This leads to a slight decrease in contrail cirrus coverage and radiative forcing in the tropical areas (by ~5%) and over Eastern China/Japan (by ~20%) (Fig. 4b and d). The contrail cirrus cover decreases in the Eastern China/Japan

region (Fig. 4d) due to a lower ice supersaturation frequency and less frequent formation of persistent contrails. This leads to a decrease in radiative forcing (Fig. 4b) and in radiative forcing per flight distance (Fig. 3c) over Eastern China/Japan. Over Europe and the U.S./Mexico contrail cirrus coverage and optical depth is slightly increased (by ~5%). Our simulations show that contrail cirrus optical depth increases over the U.S./Mexico (Fig. 4d,e and f), which leads to a slight increase in contrail



cirrus radiative forcing over Europe and the U.S./Mexico (Fig. 4b). These two different effects, an increase of contrail cirrus radiative forcing over the U.S./Mexico and over Europe and a decrease over Eastern China/Japan and the tropical areas, almost compensate each other (Fig. 4b).

### 3.4 Reduced soot emission and improvement in propulsion efficiency

A reduction in the initial contrail ice particle number by 50% leads to a strong decrease in the climate impact of contrail cirrus reducing global radiative forcing for the year 2050 by 14% from 160 to 138 mWm$^{-2}$ (Table 1). A smaller number of initial ice crystals can grow faster assuming a constant amount of ambient water vapor available for condensation, leading to an earlier and larger sedimentation loss of ice crystals (Bier et al., 2017) and therefore, to a decrease in contrail cirrus optical depth, life times and radiative forcing (Burkhardt et al., 2018). The decrease in contrail cirrus radiative forcing for the year

2050 is caused by a decrease in contrail cirrus optical depth of up to 30% (Fig. 4 e and f, Fig. 6) and a decrease in contrail cirrus coverage (Fig. 4d). The changes in radiative forcing are largest over South East Asia/India where sedimentation plays a greater role due to the larger amount of water vapor available for condensation. Over Europe the effect is slightly larger than over the U.S./Mexico, because of its location downwind of the North Atlantic flight corridor where contrail cirrus coverage is strongly influenced by the life time of contrail cirrus originating over the Atlantic. The smallest impact of the

reduction in initial ice crystal numbers on contrail cirrus radiative forcing among the four studied regions can be found over the U.S./Mexico (Fig. 4c) where contrail cirrus coverage mainly consists of young contrails.

The impact of soot reductions is smaller than estimated in Burkhardt et al. (2018) who found that a 50% reduction in soot emissions causes a 20% reduction in contrail cirrus radiative forcing for air traffic of the year 2006. The difference in sensitivity may be caused by the change in air traffic volume and pattern. Contrail cirrus radiative forcing is nonlinearly

dependent on the initial ice crystal number (Burkhardt et al., 2018). This means that reducing initial ice crystal numbers in an increased air traffic environment has a smaller impact on contrail cirrus radiative forcing than for current air traffic since an abundance of contrail cirrus ice crystals will still exist even if nucleation rates are reduced.

The increase in propulsion efficiency and the change in water vapor emissions (Sect. 2.3) has no significant impact on contrail cirrus radiative forcing. Only in the tropics contrail formation probability around 250 hPa is slightly increased (Fig.

5b), which has no significant impact on the global radiative forcing due to contrail cirrus.

### 4. Discussion

Only one study exists that analyzes the impact of contrail cirrus on the radiative balance in the future and another study looks at the change in line-shaped contrails only. Chen and Gettleman (2016) use a very different approach simulating contrail

cirrus, calculating the number of newly formed contrail ice crystals from the available water vapor, setting the size of the ice crystals constant, and feeding this tendency in ice crystal number into the natural cloud scheme. Their resulting estimate of contrail cirrus radiative forcing for the year 2006, 13 mWm$^{-2}$, is significantly smaller than our estimate which is likely connected with an underestimation of ice crystals formed at contrail formation resulting from assumed ice crystals sizes



larger than observed in young contrails (Schumann and Heymsfield, 2017). Due to the 4 fold increase in air traffic they estimate an increase in contrail cirrus radiative forcing by a factor of 7, which they argue is caused by non-uniform increases in air traffic and regional differences in sensitivity to air traffic. We calculate a 3 fold increase connected with the 4 fold increase in air traffic which is in line with the 3 fold increase in contrail cirrus coverage predicted by our model. Finally

Chen and Gettelman (2016) estimate a decrease of contrail cirrus radiative forcing by about 12% and 8% assuming RCP 8.5 and RCP 4.5, respectively, whereas we find that changes in contrail cirrus radiative forcing due the changing climate (assuming RCP 6.0) cancel out globally. This difference in the impact of climate change on contrail cirrus radiative forcing is caused by differences in the estimated of the change in the contrail formation frequency. Whereas the decrease in contrail formation probability in the tropics is captured by both models, in the extratropics we find an increase in contrail formation

frequency which lies further to the south than in the simulations of Chen and Gettelman (2016, their Fig. 2) and thus in our simulation still effects contrail formation over the U.S./Mexico region. This leads in our study to a cancellation of the decrease in contrail formation in the tropics and an increase in the extratropics due to climate change. Future changes in contrail cirrus properties and radiative forcing due to a changing climate are much more uncertain in the mid-latitudes than in the tropics since the trend in ice supersaturation frequency in the mid-latitudes is strongly model dependent (Irvine and

Shine, 2015).

Marquart et al (2003), who study only line-shaped contrails, use an approach that relies on the scaling of the contrail formation probability over a specified area to observations, show a strong decrease of line-shaped contrail coverage in the tropics due to climate change by up to 70%. Their method is connected with a number of weaknesses, firstly an error in the parameterization of potential contrail coverage which is effective especially in the tropics (Burkhardt et al., 2008),

assumptions about the scalability of contrail cirrus coverage (Burkhardt et al., 2010) that assume contrail cirrus life cycles to be equal in the extratropics and tropics which is not justified (Burkhardt et al., 2018) and the assumption that scaling coefficients can be transferred from our to a future climate.

However, all studies agree that increasing air traffic is the dominating effect that causes higher contrail cirrus radiative forcing in the future and the Chen and Gettelman study and our study agrees on the change in climate having only a small

impact on contrail cirrus radiative forcing.

## 5. Conclusion

In this paper, we present contrail cirrus properties and radiative forcing for the year 2050 using AEDT emission scenarios. We isolate effects that can be expected from the change in air traffic volume and its geographic and vertical distribution, from climate change, from improvements in fuel and propulsion efficiency and decreases in soot and water vapor emissions,

caused by the use of alternative fuels. We study regional changes in the main air traffic areas and in areas where air traffic is projected to increase strongly.





We find that the future projected increase in air traffic and the slight shift to higher altitudes leads to a large increase of contrail cirrus coverage, optical depth and radiative forcing. With a four-fold increase in air traffic contrail cirrus radiative forcing is increasing three-fold, from 49 to 159 mWm$^{-2}$. The main air traffic areas over Northern America and Europe continue to contribute the largest fraction of the contrail cirrus radiative forcing but the Asian main air traffic areas gain in

importance. Our estimates of current and future contrail cirrus radiative forcing are different to those given by Chen and Gettelman (2016) which is likely connected with their methodology estimating contrail ice nucleation (see Sect. 5). Contrail cirrus radiative forcing appears to be hardly affected by climate change assuming RCP 6.0, which leads to a slight decrease in contrail cirrus coverage and radiative forcing over Asia and a compensating small increase over Northern America and Europe. This is in contrast to results from Chen and Gettelman (2016) who found contrail cirrus radiative forcing to decrease

due to climate change by about 12% assuming RCP 8.5. The reason for this discrepancy can be traced back to a difference in the pattern of change of contrail formation probability in the northern hemisphere. The maximum increase in contrail formation probability lies within the midlatitudes whereas in the Chen and Gettelman (2016) simulation it is found north of our maximum. Nevertheless, the studies agree that changes in contrail cirrus radiative forcing due to the projected increase in air traffic by far outweigh any damping effect that a change in climate may have.

Of the fuel and propulsion efficiency improvements and soot reductions due to the use of alternative fuels, it is the soot reduction that has the largest impact. The larger propulsion efficiency leads to a slight increase in the contrail formation probability in the tropics with little impact on global radiative forcing. The soot emissions cause a reduction in contrail cirrus optical depth and life time (Burkhardt et al., 2018) which leads again to a decrease in contrail cirrus coverage. Consequently contrail cirrus radiative forcing is decreased by 15%, less than estimated by Burkhardt et al. (2018) who infer a 20%

reduction for air traffic in the year 2006. This slight decrease in sensitivity connected with soot number emission reductions is likely caused by the fact that the strong increase in air traffic leads to an abundance of ice crystals which makes decreases in ice crystal numbers less effective.

Overall the strong increase in radiative forcing from 2006 to 2050 due to larger air traffic volume and the shift of air traffic towards higher altitudes cannot be compensated by small reductions in radiative forcing due to changes expected from

climate change, reduced soot emissions and improvements in fuel efficiency.

Contrail cirrus radiative forcing per flight distance appears to be particularly high in the tropics. This result should still be viewed with some caution, since in the tropical areas contrails form close to the threshold conditions which should lead to a lower contrail ice crystal nucleation rate (Kärcher et al., 2015). This has implications not only for contrail optical depth but also for the ice crystal loss rates during the vortex phase (Unterstrasser, 2016), microphysical process rates (Bier et al., 2017)

and contrail cirrus life times (Burkhardt et al., 2018). When including a parameterization for contrail ice crystal nucleation this is likely to lead to a decrease in contrail cirrus radiative forcing in the tropics. The impact of the tropical areas on global contrail cirrus radiative forcing is still very limited so that the overestimation of contrail cirrus ice crystals has a limited impact on global contrail cirrus radiative forcing. As air traffic increases strongly in the tropical areas future simulations



should include the impact of lower nucleation rates and the associated changes in ice crystal loss rates, changes in optical depth, microphysical process rates and contrail cirrus life time in the tropics.

*Data avaibility.* The data obtained from this study are available upon request from the authors.

*Author contributions.* L. Bock perfomed and analysed simulations. L.Bock and U. Burkhardt jointly discussed scientific
10   results and wrote the paper.

*Competing interests.* The authors declare that they have no conflict of interest.

*Acknowledgements.* The authors thank Michael Ponater for helpful comments, the Volpe National Transportation Center and the U.S. Federal Aviation Administration and C.-C. Chen for providing the AEDT inventories. The work was funded by a postdoc program of Rolf Henke, member of the DLR executive board. The model simulations were performed at the German Climate Computing Centre (DKRZ) through support from the Bundesministerium für Bildung und Forschung (BMBF).

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



| Simulation | background climate | inventory | air traffic volume [km a$^{-1}$] | propulsion efficiency | initial ice number concentration [cm-3] | coverage [%] | RF [mWm$^{-2}$] |
|---|---|---|---|---|---|---|---|
| 2006 | 2006 | 2006 | $3.7 * 10^{10}$ | 0.3 | 150 | 1.1 (0.7) | 49 |
| 2006 Plus | 2006 | 2050 Baseline | $15.4 * 10^{10}$ | 0.3 | 150 | 2.9 (2.0) | 159 |
| 2050 Baseline | 2050 (RCP 6.0) | 2050 Baseline | $15.4 * 10^{10}$ | 0.3 | 150 | 2.8 (2.0) | 160 |
| 2050 Scenario 1 | 2050 (RCP 6.0) | 2050 Scenario 1 | $15.4 * 10^{10}$ | 0.42 | 75 | 2.8 (1.7) | 137 |

**Table 1: Overview over the model simulations. Air traffic distance is given as ground projected track distance. Coverage is given for all contrail cirrus and in brackets for visible (optical depth > 0.05) contrail cirrus only (Bock and Burkhardt, 2016b).**



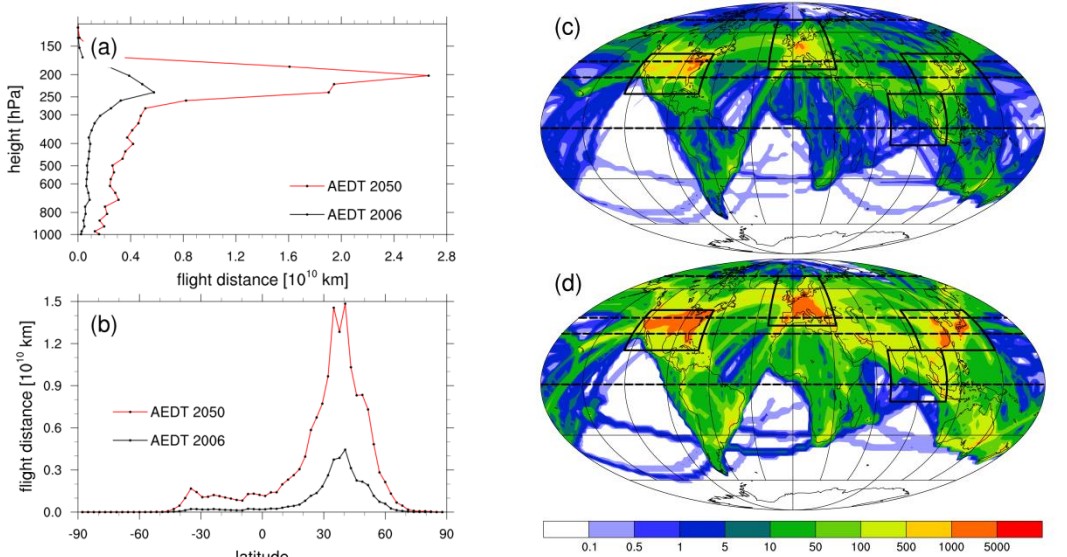

**Figure 1: Vertical (a) and zonal (b) distribution of total annual flight distance and (c) horizontal distribution of vertically integrated air traffic density [km m⁻² s⁻¹] for the years (c) 2006 and (d) 2050.**





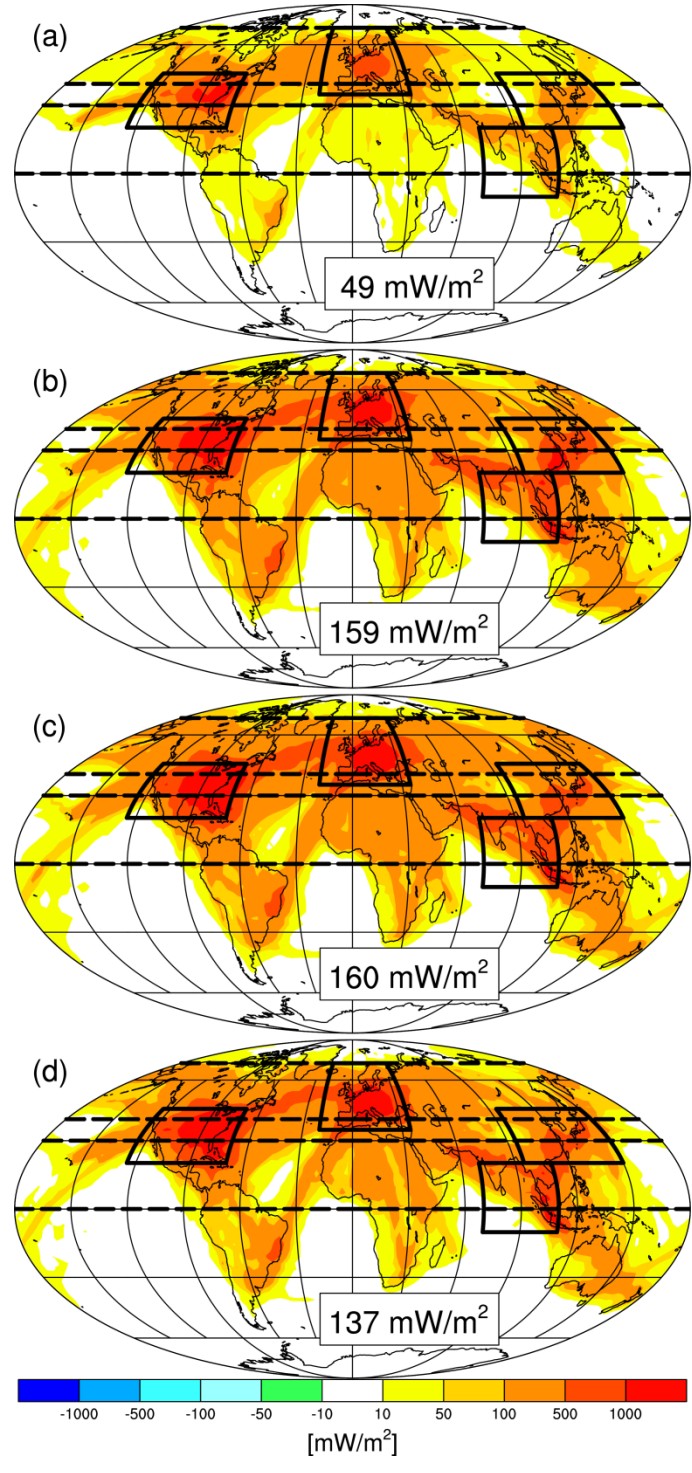

**Figure 2: Radiative forcing in scenarios 2006 (a), 2006 Plus (b), 2050 Baseline (c) and 2050 Scenario 1 (d). Boxes (solid lines) and latitude bands (dashed lines) indicate regions (defined in Sect 2.3) which we compare in Fig. 3 and 4.**



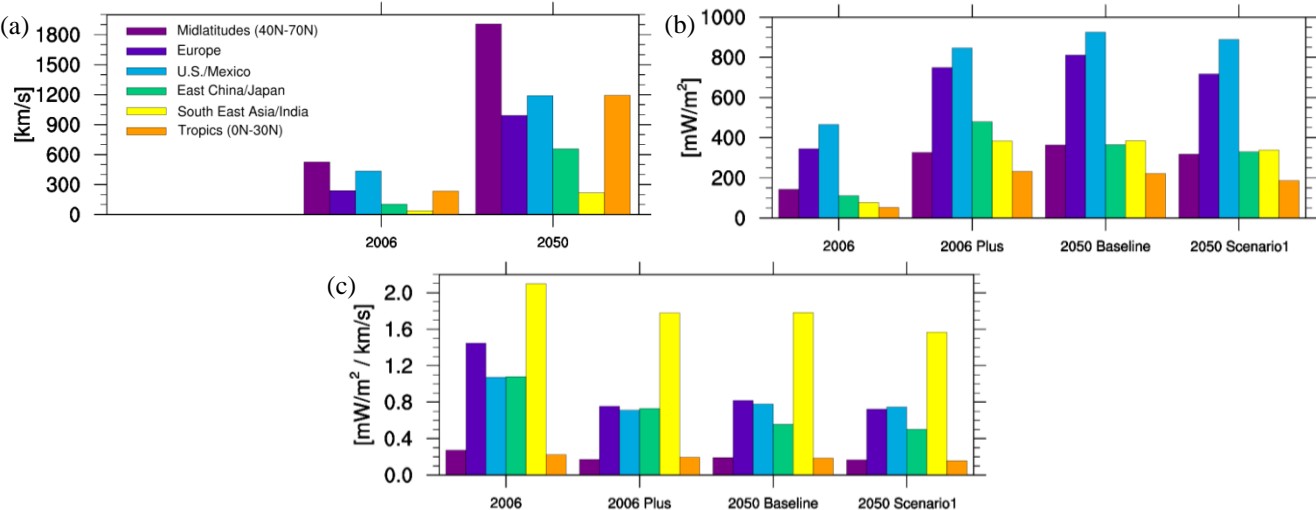

**Figure 3: a) Flight distance [10^10 km] per year for 2006 and 2050 and b) contrail cirrus radiative forcing [mWm^-2] and c) contrail cirrus radiative forcing per flight distance for simulations summarized in Table 1 in different regions (same area size), in the mid-latitudes and in the tropics.**





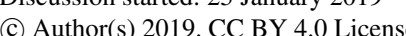

**Figure 4: Factor of change for flight distance, radiative forcing (RF) and ratio RF over flight distance from scenario 2006 to 2006P (a), 2006P – 2050 B (b) and 2050 B to 2050 S1 (c). Absolute change of (d) mean coverage (in %) due to contrail cirrus with an optical depth of >0.05 and (e) mean optical depth at 200 hPa and (f) 240 hPa for different areas.**





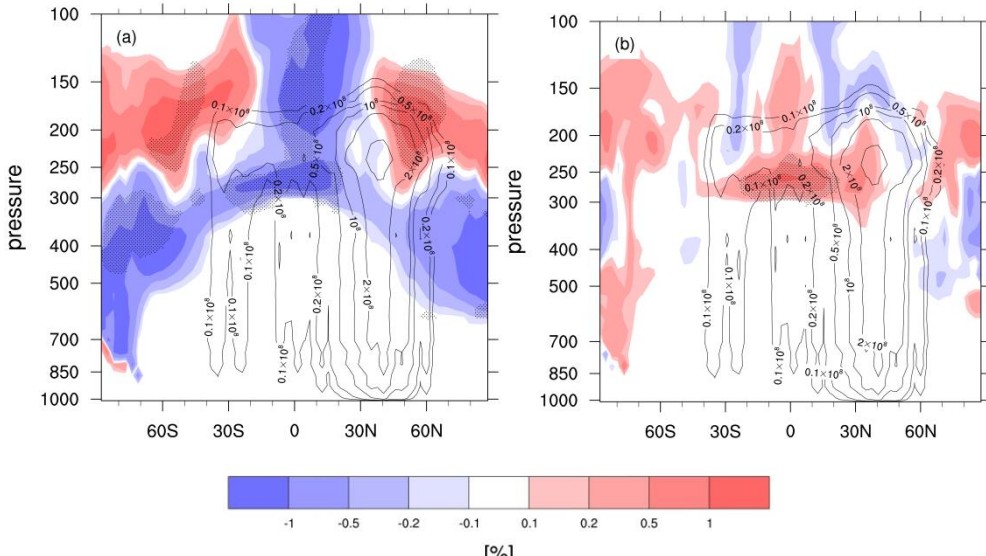

**Figure 5: Changes in contrail formation frequency from 2006 to 2050 due to climate change (a) and due to improved propulsion efficiency (b). Contour lines indicate annual flight distance [km] in 2050. Dotted regions indicate statistically significant changes.**





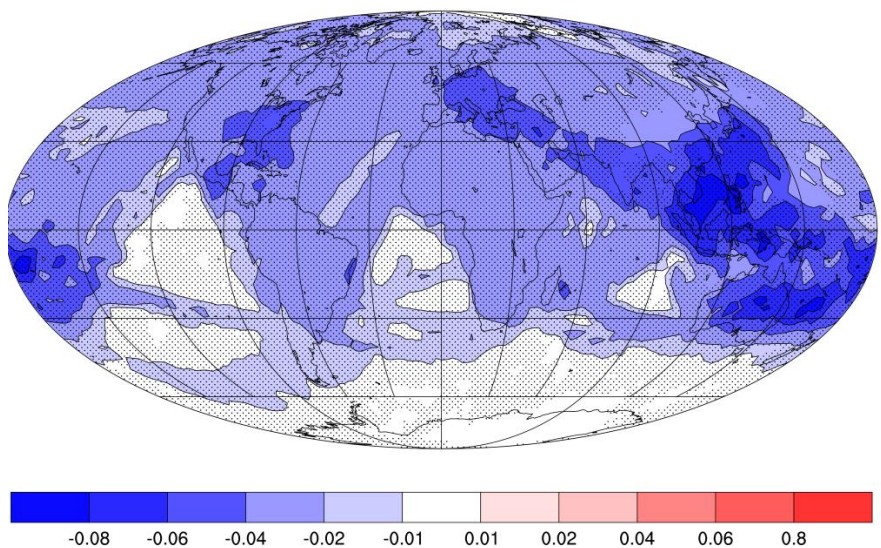

**Figure 6: Absolute difference in optical depth in 200hPa between scenario 2050 B and 2050 S1 due to soot reductions. Dotted regions are significant.**