# Peer review of "Contrail cirrus radiative forcing for future air traffic"

_Atmospheric Chemistry and Physics, 2018_

## Referee Comment (RC1) · Anonymous Referee #1 · 12 Mar 2019

General Comments

The article estimates the increase in contrail cirrus radiative forcing (RF) between 2006 and 2050, separating the contributions from the increase in air traffic and cruise altitude, reduction in soot emissions, and background meteorology differences linked to future climate change. The results report an RF increase by a factor of 3 with a relatively modest reduction (15%) from a 50% soot number emissions decrease, concluding that the increase in RF linked to traffic growth (factor of 4) cannot be counterbalanced by improvements in propulsion efficiency and soot emissions.

The manuscript addresses a relevant topic, providing guidance for attribution and mitigation options for the contribution of the aviation sector to climate change. The methodology is sound and based on a tested model and all the sections of the manuscript are

clearly presented in a logical way. The manuscript definitely fulfil ACP's standards, and is unreservedly recommended for publication. Only very minor suggestions are made that I hope will improve the clarity and interpretation of the results.

Specific suggestions:

Pg 4 ln 22: It would probably be useful to expand on the magnitude of the future flight altitude shift and add a reference. This will enable the reader to get a sense of the sensitivity of your model to such changes.

Pg 6 ln 19: add a comma after the word "large".

Pg 6 ln 19: It would probably be useful to expand, in the sentence starting in ln 19, if the reduction in ice crystal numbers in the tropics is accounted for in the parameterisation.

Pg 9 ln 8: remove "of the"

Pg 9 ln 16: add the period in "et al"

---

## Referee Comment (RC2) · Anonymous Referee #2 · 14 Mar 2019

This is a very interesting and generally clearly written paper, in an important area where there have been rather few earlier papers in the literature. I recommend it be accepted following modifications.

The more important comments are preceded by an "M".

1:12 It would be useful to say in the abstract what the 2050 forcing is, in W/sq.m, rather than just reporting the 2050:2006 ratio (especially as this paper disagrees significantly with the only other recent study).

1:16 The authors may have been up against a word limit here, but I feel it would be useful to qualify the statement on the global-mean insignificance of the climate change by a comment that there are regional forcing differences resulting from the effect of

climate change.

2:19 This crops up a few times in the manuscript. Is it correct that it is "a lower number OF LARGER ice crystals"? If so, I think this would add clarity.

2:33 Another frequent (albeit minor) issue. Optical depth is a wavelength-dependent quantity. I presume the authors mean "visible"? This should be clarified.

3:9 "different sensitivities" - different sensitivities to what? Changes in air traffic volume?

M4:10 I realise that the authors may push back on this suggestion, but given the likely policy interest in the results from this paper, it may be useful to provide a couple of sentences on how the AEDT scenarios are compliant, or not, with CORSIA ("stabilise CO2 emissions by 2020 and reduce 2005 emissions by 50% by 2050"), given that the CORSIA agreement came after these scenarios were developed. I realise that this is not simple, given the role that offsetting might play in meeting the CORSIA targets. Perhaps there is a catch-all paper on the implications of CORSIA on future emission scenarios in the aviation literature that can be referred to?

4:22 – 4:23 What is the vertical resolution of the AEDT dataset? Is the change from 200 to 240 hPa a shift in one level in the dataset (in which case, the interpretation requires some caution) or is it several levels, and hence more robust. Or perhaps this is the resolution of the climate model, rather than the parent dataset? From Fig 1a it looks as though it may be two levels, but still it is unclear whether that is the climate model or dataset resolution.

5:5 Presumably there is a slight inconsistency here in that the HadGEM SSTs would have been forced by more than just changes in CO2?

M4:28 – 5:3 I didn't find the simulation names very intuitive, especially 2006P when it is really 2050, and this inhibited understanding of the paper. I wonder if something like 2050T (T for traffic), 2050TC (C for climate), 2050TCS (S for soot or maybe M for

ACPD
modified efficiency and fuels) would help the reader more?

M6:4-6:8 This should be explained more clearly. I eventually understood that "slant" meant in the vertical rather than horizontal dimension and that "track distance" and "ground projected" were the same thing. The paper does not clearly say that "slant" is better, but this is what I assume. That led me to wonder whether the global estimates in this paper should be multiplied by the factor of 1.14 to give a more reliable answer.

6:21-6:22 I found this unclear, even having worked in the area, and suggest the text is expanded to make it clearer. Is the "formation threshold" referring to a temperature or supersaturation threshold or both?

6:23 and 6:26: Sentences repeated? Also point out to the reader (6:23) that the ice supersaturation is not shown?

M7:5 The "shift of a large fraction" is interesting/ important but too vague. Could this be made quantitative? Presumably it differs between summer and winter, as the tropopause itself changes so much in mid-latitudes. Or perhaps this has been discussed in another paper and a reference could be given?

7:26 Would "contrail formation frequency" be better described as the "probability of contrail formation"? The frequency is dependent on an aircraft flying through the relevant grid box and so could be zero even if the probability is 1.

9:9 NORTHERN extratropics

9:14 Strictly I think this is the cold ice supersaturation frequency – as I understand it, it is the warming, rather than the humidity change, that is most influential in changing the tropics

9:25 Add "global-mean" to this sentence.

M10.3 If estimates of aviation CO2 radiative forcing from the 2 AEDT scenarios (or the CO2 amounts - as the forcing could be derived from the simple IPCC expressions)
are easily available from other papers, their addition here (and the relative growth from 2006) would be useful to place the growth of the contrail forcing in perspective. It would be particularly useful to know if the contrail forcing grows more/less rapidly than the CO2 forcing. This may need a further caveat given the Ponater et al. (Atmos Env 2006) and Rap et al. (GRL 2010) papers indicating that the efficacy of contrail forcing may be significantly less than 1.

Figure 1: In (a) the (a) label obscures the lines. Also the y-axis is pressure not height

Figure 2 (a) caption says km per year but the y-axis label says km per second. I note that the labels (US/Mexico and East China/Japan) differ between here and Fig 4, and the text itself. I suggest making consistent.

M:Figure 3 needs some work to help the reader. On initial viewing it is indigestible. Yaxis labels are missing, when they need not be, and their addition would make it much clearer. It is also here that I most felt a more intuitive use of simulation names would help the reader. "2006 plus" feels particularly unhelpful.

Figure 5: The power of 10 labels are unreadable to me. Could they be removed from the figure and included in the caption instead?

Typos etc

1:26 "live time" -> "lifetime"

2:28 Irvine et al. missing from reference list, unless this meant to be Irvine and Shine

4:11 and 11:23 Barret -> Barrett

5:11 I advise using x not \* for multiplying factors of 10 - see also Table 1 (maybe irrelevant if dealt with at the typesetting stage)

5:24 - 5:26 The O's are for Ost?

10:6 Section 4 not 5?

---

## Author Comment (AC1) · 17 May 2019

**Answer to Reviewer #1:**

**The article estimates the increase in contrail cirrus radiative forcing (RF) between 2006 and 2050, separating the contributions from the increase in air traffic and cruise altitude, reduction in soot emissions, and background meteorology differences linked to future climate change. The results report an RF increase by a factor of 3 with a relatively modest reduction (15%) from a 50% soot number emissions decrease, concluding that the increase in RF linked to traffic growth (factor of 4) cannot be counterbalanced by improvements in propulsion efficiency and soot emissions.**

**The manuscript addresses a relevant topic, providing guidance for attribution and mitigation options for the contribution of the aviation sector to climate change. The methodology is sound and based on a tested model and all the sections of the manuscript are clearly presented in a logical way. The manuscript definitely fulfils ACP's standards, and is unreservedly recommended for publication. Only very minor suggestions are made that I hope will improve the clarity and interpretation of the results.**

Thank you for your very positive judgement and for comments.

**Specific suggestions:**

**Pg 4 ln 22: It would probably be useful to expand on the magnitude of the future flight altitude shift and add a reference. This will enable the reader to get a sense of the sensitivity of your model to such changes.**

This is a very good point but unfortunately the information on the shift that we were able to acquire is limited. We received the data set horizontally gridded and with relatively low vertical resolution (30 levels) from C.-C. Chen, NCAR. We know that the shift causes the maximum of air traffic distance in 2050 to be located at 200 hPa instead of 240 hPa as in the old inventory. Additionally M. Gupta from the FAA assured us that this shift of flight altitude is a realistic consumption in the Volpe future scenario and wrote us that new aircraft seem to fly at higher altitude than the old ones with a difference ranging between 0.3-1.5km. The shift from 200 to 240 hPa lies with about 1 km in the middle of this range.

We added "by between 0.3 and 1.5 kilometres (pers. comm. Mohan Gupta, FAA), resulting in the shift of maximum flight density seen in Fig. 1a" .

**Pg 6 ln 19: add a comma after the word "large".**

Thanks.

**Pg 6 ln 19: It would probably be useful to expand, in the sentence starting in ln 19, if the reduction in ice crystal numbers in the tropics is accounted for in the parameterisation.**

The reduction of initial ice crystal numbers is not considered in our parameterization as we describe a constant initial ice crystal concentration for contrail.

We modified for clarification this part: "It needs to be pointed out that contrail optical depth is likely overestimated in the tropics, since in the tropics contrails form within a few degrees of the temperature threshold (Schmidt-Appleman criterion) limiting ice nucleation in the contrail (Bier and Burkhardt, 2019), a process that is not resolved in our simulations (Sect. 2.3). Therefore optical depth and lifetimes of contrails will be overestimated (Burkhardt et al., 2018) and consequently radiative forcing."

**Pg 9 ln 8: remove "of the"**

Thanks.

**Pg 9 ln 16: add the period in "et al"**

Thanks.

---

## Author Comment (AC2) · 17 May 2019

**Answer to Reviewer #2:**

**This is a very interesting and generally clearly written paper, in an important area where there have been rather few earlier papers in the literature. I recommend it be accepted following modifications.**

**The more important comments are preceded by an "M".**

Thank you for your in-depth review that helped us to improve our paper.

**1:12 It would be useful to say in the abstract what the 2050 forcing is, in W/sq.m, rather than just reporting the 2050:2006 ratio (especially as this paper disagrees significantly with the only other recent study).**

Yes, we added the absolute number of the 2050 contrail cirrus radiative forcing.

**1:16 The authors may have been up against a word limit here, but I feel it would be useful to qualify the statement on the global-mean insignificance of the climate change by a comment that there are regional forcing differences resulting from the effect of climate change.**

Yes, we added this point.

**2:19 This crops up a few times in the manuscript. Is it correct that it is "a lower number OF LARGER ice crystals"? If so, I think this would add clarity.**

As we wrote here about newly nucleated ice crystals, they may not or only very slightly be larger than when forming more ice crystals. But when deposition starts, then these fewer crystals could grow larger.

We added "with ice crystals grow to larger sizes and sedimentation is initiated earlier in the life cycle.".

**2:33 Another frequent (albeit minor) issue. Optical depth is a wavelength-dependent quantity. I presume the authors mean "visible"? This should be clarified.**

Yes, we added "visible" in a few locations.

**3:9 "different sensitivities" – different sensitivities to what? Changes in air traffic volume?**

To clarify this sentence we add: "to an increased air traffic volume".

**M4:10 I realise that the authors may push back on this suggestion, but given the likely policy interest in the results from this paper, it may be useful to provide a couple of sentences on how the AEDT scenarios are compliant, or not, with CORSIA ("stabilize CO2 emissions by 2020 and reduce 2005 emissions by 50% by 2050"), given that the CORSIA agreement came after these scenarios were developed. I realise that this is not simple, given the role that offsetting might play in meeting the CORSIA targets. Perhaps there is a catch-all paper on the implications of CORSIA on future emission scenarios in the aviation literature that can be referred to?**

This is a very interesting connection.

The AEDT 2050 Baseline describes unconstrained growth with a factor of 4.8 increase in fuel burn relative to the year 2006. The 2050 Scenario1 includes an improvement of fuel efficiency by 2% per year and additional $NO_x$ emission reductions. The CORSIA aims, to stabilize net $CO_2$ emissions from international civil aviation at 2020 levels and to reduce the net $CO_2$ emissions to half of what they were in 2005, by 2050, would require a much larger reduction in fuel burn or significant carbon offsetting. It is important to realize that while carbon offsetting may be helpful in order to limit the impact of $CO_2$ emissions the large impact of contrail cirrus is unaffected by carbon offsetting.

We added: "As fuel burn increases by a factor of 4.8 between 2006 to 2050 Baseline and still by a factor of 2.7 between 2006 and 2050 Scenario 1 (Unger et al., 2013), the specifications of the future projections do not meet the requirements of the international CORSIA agreement unless the remaining necessary $CO_2$ emission reductions are introduced purely by carbon offsetting.

**4:22 – 4:23 What is the vertical resolution of the AEDT dataset? Is the change from 200 to 240 hPa a shift in one level in the dataset (in which case, the interpretation requires some caution) or is it several levels, and hence more robust. Or perhaps this is the resolution of the climate model, rather than the parent dataset? From Fig 1a it looks as though it may be two levels, but still it is unclear whether that is the climate model or dataset resolution.**

This is a very good point. Mohan Gupta from FAA assured us that the shift of flight altitude in the Volpe future scenario is realistic and caused by the fact that new aircraft are expected to fly at a higher altitude than the older ones with the height difference ranging between 0.3-1.5km. We received the inventory horizontally and vertically gridded (with 30 vertical levels) from C.-C. Chen, NCAR, and find a vertical shift of air traffic, causing the maximum of air traffic distance in 2050 to be located at 200 hPa instead of 240 hPa as in the inventory for 2006. This shift from 200 to 240 hPa amounts to about 1 km height and is therefore consistent with the shift of between 0.3 and 1.5 km.

We added "The AEDT flight inventory that we use in our model has originally a 1°x1° horizontal resolution with 30 vertical levels, transformed by recursive conservative mapping (Jöckel, 2006) to our model resolution." and "… shift upwards by between 0.3 and 1.5 kilometers (pers. comm. M. Gupta, FAA), resulting in the shift of maximum flight density seen in Fig. 1a.".

**5:5 Presumably there is a slight inconsistency here in that the HadGEM SSTs would have been forced by more than just changes in CO2?**

Yes, this is true. We considered here only the change in $CO_2$ and SST as they are responsible for a major part of the change in the upper troposphere and lower stratosphere, the area where contrails form.

We added in "Other than that emissions and boundary data are not changed."

**M4:28 – 5:3 I didn't find the simulation names very intuitive, especially 2006P when it is really 2050, and this inhibited understanding of the paper. I wonder if something like 2050T (T for traffic), 2050TC (C for climate), 2050TCS (S for soot or maybe M for modified efficiency and fuels) would help the reader more?**

The year in the scenario name indicates the climate state in the model. We changed scenario names in order to make them more intuitive. They are now C2006-T06, C2006-T50, C2050-T50 and C2050-T50M (with "C" for climate, "T" for traffic and "M" for mitigation).

**M6:4 – 6:8 This should be explained more clearly. I eventually understood that "slant" meant in the vertical rather than horizontal dimension and that "track distance" and "ground projected" were the same thing. The paper does not clearly say that "slant" is better, but this is what I assume. That led me to wonder whether the global estimates in this paper should be multiplied by the factor of 1.14 to give a more reliable answer.**

Yes, the slant distance is the more realistic one, as we could initialize the contrails in a model gridbox with the exact length and volume, for explanation and discussion see Sect. 2.2. Unfortunately we do not have the slant distance numbers for the 2050 scenario. When we assume that the factor from track distance to slant distance stays constant in future scenarios we can apply it to the result we got for the 2050 scenario.

We repeated in Sect. 3.1 the definition of track and slant distance and added: "The results are based on an air traffic inventory of future air traffic measured as track (ground projected) distance rather than slant (3D) distance. Assuming that the relation of contrail cirrus radiative forcing calculated from track or slant distance stays constant for future scenarios and therefor applying the factor 1.14 (Bock and Burkhardt, 2016b), this would correspond to a global mean contrail cirrus radiative forcing of 182 mWm$^{-2}$ that would be resulting from an inventory of future air traffic measured in slant distance. " We added this rounded value in the abstract and the extrapolated values for all scenarios in the Tab. 1.

**6:21-6:22 I found this unclear, even having worked in the area, and suggest the text is expanded to make it clearer. Is the "formation threshold" referring to a temperature or supersaturation threshold or both?**

As we only study persistent contrails forming within ice supersaturated air, the formation criterion (Schmidt-Appleman criterion) is a temperature criterion.

We modified this part: "It needs to be pointed out that contrail optical depth is likely overestimated in the tropics, since in the tropics contrails form within a few degrees of the temperature threshold (Schmidt-Appleman criterion) limiting ice nucleation in the contrail (Bier and Burkhardt, 2019), a process that is not resolved in our simulations (Sect. 2.3). Therefore optical depth and lifetimes of contrails will be overestimated (Burkhardt et al., 2018) and consequently radiative forcing."

**6:23 and 6:26: Sentences repeated? Also point out to the reader (6:23) that the ice supersaturation is not shown?**

Thanks. We deleted the second sentence and added "(not shown)" in the text.

**M7:5 The "shift of a large fraction" is interesting/ important but too vague. Could this be made quantitative? Presumably it differs between summer and winter, as the tropopause itself changes so much in mid-latitudes. Or perhaps this has been discussed in another paper and a reference could be given?**

In the absence of a diagnostic in our model output giving us the shift into the stratosphere we calculate the shift relative to an annually mean tropopause. The fraction of air traffic above the mean tropopause in the midlatitudes (40 to 50°N) increases from 16 to 29% from 2006 to 2050. We added this information: "(in the northern midlatitudes the fraction increases on average from 16 to 29%)".

**7:26 Would "contrail formation frequency" be better described as the "probability of contrail formation"? The frequency is dependent on an aircraft flying through the relevant grid box and so could be zero even if the probability is 1.**

We corrected the figure caption of Fig. 5 to "persistent contrail formation probability" and changed it in the text.

**9:9 NORTHERN extratropics**

Thanks.

**9:14 Strictly I think this is the cold ice supersaturation frequency – as I understand it, it is the warming, rather than the humidity change, that is most influential in changing the tropics**

We agree that the main reason for the changes in the tropics is the warming and we tried to clarify this statement with adding "caused mainly by temperature changes " to the text.

**9:25 Add "global-mean" to this sentence.**

Yes, we did.

**M10.3 If estimates of aviation CO2 radiative forcing from the 2 AEDT scenarios (or the CO2 amounts - as the forcing could be derived from the simple IPCC expresssions) are easily available from other papers, their addition here (and the relative growth from 2006) would be useful to place the growth of the contrail forcing in perspective. It would be particularly useful to know if the contrail forcing grows more/less rapidly than the CO2 forcing. This may need a further caveat given the Ponater et al. (Atmos Env 2006) and Rap et al. (GRL 2010) papers indicating that the efficacy of contrail forcing may be significantly less than 1.**

To include this interesting point in our paper we added a two paragraphs to the conclusion:

"In order to understand the implications of our results for the overall air traffic climate impact, we calculated the aviation $CO_2$ radiative forcing according to Myhre et al. (1998). $CO_2$ emissions and contrail cirrus radiative forcing are the two largest aviation related radiative forcing components besides the possibly large but as yet unquantified impact of indirect effects on clouds (Lee et al., 2009). Radiative forcing due to aviation $CO_2$ emissions amounts for 2006 to 24.0 mWm$^{-2}$ and for the year 2050, assuming the C2050-T50 scenario, to 84.8 mWm$^{-2}$ and assuming the C2050-T50M scenario to 58.0 mWm$^{-2}$. This means that the factor of increase in $CO_2$ radiative forcing from C2006-T06 to C2050-T50 is 3.5, slightly higher than 3.2 for the contrail cirrus radiative forcing. Considering the increase in fuel efficiency from C2006-T06 to C2050-T50M, the factor of change for the $CO_2$ radiative forcing is reduced to 2.4, whereas the factor of change for the global contrail cirrus radiative forcing in this scenario is reduced to 2.8. The decrease in contrail cirrus radiative forcing in this scenario is caused by the decrease in soot emissions. This means that radiative forcing due to contrail cirrus can be expected to increase faster in the future than that due to $CO_2$.

The increase in fuel efficiency included in the AEDT inventory does not conform with the CORSIA agreement unless a large part of the $CO_2$ emission reduction is reached by carbon offsetting. It is important to point out that carbon offsetting deals only with the impact of $CO_2$ emissions while leaving the impact of contrail cirrus on climate unchanged. Since the increase in contrail cirrus radiative forcing can be stronger than in $CO_2$ radiative forcing both radiative forcing components need to be considered in future agreements."

As we calculate contrail cirrus radiative forcing and give no implications to surface temperature, we do not want to open the discussion about efficacy of contrail cirrus in this paper.

**Figure 1: In (a) the (a) label obscures the lines. Also the y-axis is pressure not height**

Thanks. We modified the figure.

**Figure 2 (a) caption says km per year but the y-axis label says km per second. I note that the labels (US/Mexico and East China/Japan) differ between here and Fig 4, and the text itself. I suggest making consistent.**

I suppose you meant Fig. 3. Indeed we show the flight distance in km/s, therefor we changed the figure caption. Regarding the regions, we updated Fig. 4 with the consistent names.

**M:Figure 3 needs some work to help the reader. On initial viewing it is indigestible. Yaxis labels are missing, when they need not be, and their addition would make it much clearer. It is also here that I most felt a more intuitive use of simulation names would help the reader. "2006 plus" feels particularly unhelpful.**

We hope we could clarify more with adding the y-axis labels and renaming the scenarios (see earlier comment).

**Figure 5: The power of 10 labels are unreadable to me. Could they be removed from the figure and included in the caption instead?**

Thanks. We did.

**Typos etc**

**1:26 "live time" -> "lifetime"**

Thanks.

**2:28 Irvine et al. missing from reference list, unless this meant to be Irvine and Shine**

Thanks.

**4:11 and 11:23 Barret -> Barrett**

Thanks.

**5:11 I advise using x not * for multiplying factors of 10 - see also Table 1 (maybe irrelevant if dealt with at the typesetting stage)**

Thanks.

**5:24 – 5:26 The O's are for Ost?**

Thanks.

**10:6 Section 4 not 5?**

Thanks.